# Depression in Central and Eastern Europe: How Much It Costs? Cost of Depression in Romania

**DOI:** 10.3390/healthcare11060921

**Published:** 2023-03-22

**Authors:** Miorita Melina Iordache, Costin Octavian Sorici, Kamer Ainur Aivaz, Elena Carmen Lupu, Andrei Dumitru, Cristina Tocia, Eugen Dumitru

**Affiliations:** 1Faculty of Medicine, Ovidius University of Constanta, 1 Universitatii Alley, 900470 Constanta, Romania; 2Prof. Alexandru Obregia Psychiatry Hospital, 10 Berceni Str., 041914 Bucharest, Romania; 3Faculty of Economics, Ovidius University of Constanta, 1 Universitatii Street, 900470 Constanta, Romania; 4Faculty of Pharmacy, Ovidius University of Constanta, 900001 Constanta, Romania; 5“St. Apostol Andrew” Emergency County Hospital, 145 Tomis Blvd., 900591 Constanta, Romania; 6Research Center for the Morphological and Genetic Study in Malignant Pathology (CEDMOG), Ovidius University of Constanța, 145 Tomis Avenue, 900591 Constanta, Romania; 7Academy of Romanian Scientists, 3 Ilfov Street, 050045 Bucharest, Romania

**Keywords:** mental illness, direct cost, indirect cost, decision maker, optimization, life quality

## Abstract

Objective: The present study aims to estimate the public cost of depression in Romania during a seven-year time span to complement existing papers with data from Central and Eastern Europe and to identify and propose measures that allow efficient use of funds. Methods: We used data collected from the National Health Insurance System to analyze the main components of the cost. Findings: Indirect costs exceed direct costs. Within the direct costs, hospitalization and medicines still have an important share but are decreasing due to the intervention of outpatient services such as psychiatrists and psychotherapists. Conclusion: Since the goal is mental health, it is necessary to act early and quickly to decrease the burden in the long run. Annually, the mean direct cost of depression per patient is EUR 143 (part of it is represented by hospitalization, i.e., EUR 67, and psychotherapy, i.e., EUR 5), the mean cost of sick leaves per patient is EUR 273, and the total cost per patient is EUR 5553. Indirect costs (cost of disability and lost productive years) represent 97.17% of the total cost. An integrated approach to early diagnosis, effective treatment, monitoring, and prevention as well as included economic and social programs are needed to optimize indirect costs.

## 1. Introduction

The World Health Organization (WHO) defines mental health as a way in which people can be active and creative [1]. Among mental health disorders, depression is a global problem. Although efforts are made, there is no progress in reducing the burden since 1990 [2,3]. The European Brain Council Value of Treatment study estimated that only 52% of cases of depression are diagnosed, 62% receive treatment, 33% have good results, and another 33% have poor results. Only 12% receiving treatment referred to a psychiatrist/specialist [4]. Globally, only 30% of patients with depression receive treatment, and of these, 40% receive adequate treatment [5].

Coordination between governments, the health community, and citizens is needed [3] because “for every dollar invested in extended treatment for depression and anxiety, there is a return of $4 in better health and productivity” according to the Lancet’s mental health initiative [6].

In Europe, ESEMeD identified major depression as the most common mental disorder [7]. Data from 2021 show that depression has a high prevalence (6.38%), the highest prevalence being found in countries with high incomes. There is variability in prevalence and large differences between countries reaching the ratio 4:1 [8]. For example, in 2010 in Spain, Vieta found a prevalence of 4.75% and a public cost of EUR 3255/year/patient [9]. In 2000, Thomas found a total cost for England estimated at GBP 370 million, an increase compared to previous estimates [10]. In Germany, hospital care was the main component (43.9% of the total) [11]. In 2021, the EPICO Study showed that the societal cost of depressive disorders was estimated at EUR 6145 million. The average cost per patient/year was EUR 3402 [9,12].

For Central and Eastern Europe, studies are lacking [13]. The region has similarities from the communist period. Even if progress has been made, the development of mental health systems is ineffective [14].

The Romanian healthcare system is based on a social health insurance model in which the state’s role is significant. It allows insured persons access to a complete package of services, while uninsured persons are only entitled to a minimum set of services. In practice, only about 89% of the Romanian population was covered by the social health insurance system in 2017; approximately 17,000,000 are insured persons out of a total of 19,587,000 inhabitants. Healthcare services are provided in 41 counties and the capital Bucharest, following the norms established at the central level. The county Health Insurance Houses have contracts with medical service providers (general practitioners/family doctors, specialist doctors, laboratories, hospitals, home care providers, etc.) at the local level. In addition, healthcare providers can be paid by the Ministry of Health under national health programs [15].

The present study aims to estimate the public cost of depression in Romania during a seven year-time span, to complement existing papers with data from Central and Eastern Europe, and to identify and propose public policy measures that allow more efficient use of funds.

## 2. Materials and Methods

Our study is a retrospective analysis that used electronic data on health insurance claims at the National Health Insurance House (NHIH). All accessed data are represented by the services and their associated costs reimbursed for hospital care, outpatient care, psychotherapy, general practice, the costs of sick leaves, and the number of deaths related to depression. Additional information, i.e., the age and gender of the insured persons, was provided.

### 2.1. Sample Size and Participants

A total of 2,504,792 patient cases diagnosed and paid within the public health insurance system during 2015–2021 were used. Inclusion criteria: patients >18 years old were diagnosed in primary care, outpatient, and hospital care, and diagnosed with depression during 2015–2021. Exclusion criteria: patients <18 years old, patients who paid for consultations or medication, patients who addressed the private medical system, patients who received an incorrect diagnosis, or who did not receive treatment were not included. Other types of mood disorders were not included in the study.

The diagnosis of depression (depressive disorders) is according to the International Classification of Diseases ICD 10 [16], which includes F32 (depressive episode with three severity types (mild (F32.0), moderate (F32.1), or severe (F32.2 and F32.3), F32.8 (other depressive episode), F32.9 (depressive episode unspecified), and F33 (recurrent depressive disorder) with repeated episodes similar to the depressive episode (mild (F32.0), moderate (F32.1), or severe (F32.2 and F32.3), F33.4 (in remission), F33.8 (unspecified), and F33.9 without episodes of mania.

### 2.2. Definition of Cost Variables

Depression’s cost has been defined as the sum of *the direct costs* (DC)—the costs of resources used to treat the disease—and *indirect costs* (IC)—the number of resources lost due to the disease that can be attributed to this diagnosis [17].

### 2.3. Direct Costs

The DCs sources are the hospital, outpatient, psychotherapy, primary care, and medication. From the payer’s perspective, the DCs are real costs calculated yearly per person. The ICs include morbidity (sick leave costs—SL) and mortality (cost of productivity loss—CPL).

#### 2.3.1. Primary Care Consultations

An insured person must first contact his general practitioner doctor. The cost of this consultation is obtained by multiplying the number of consultations with the annually indexed cost point.

#### 2.3.2. Outpatient System: Psychiatric Consultations and Psychotherapy

The outpatient consultations include the consultations provided by the psychiatrists and the related service providers, counseling, and psychotherapy offered by psychotherapists for the disease codes ICD 10 F32 and F33 and were analyzed separately. Their cost is obtained by multiplying the number of consultations with the cost points given to each service and the annually indexed cost point value.

#### 2.3.3. Hospitalization

Hospitalization includes all expenses necessary for resolving individual cases, including medicines, sanitary materials, laboratory investigations, and imaging. Medical services can be provided by day hospitalization, with a maximum duration of 12 h, or by continuous hospitalization, which involves a duration of hospitalization of more than 12 h, according to the NHIH framework contract [18]. The cost of hospitalization is obtained by multiplying the number of hospitalization days with the reference cost per hospitalization day.

#### 2.3.4. Medication

The costs of medical prescriptions are given by the net cost of antidepressant drugs prescribed and reimbursed.

### 2.4. Indirect Costs

#### Morbidity Cost

Morbidity cost is represented by the total number of sick leaves (SL) given for the incapacity of working with the disease codes F32 and F33, multiplied by the working day costs specific to each insured person. The SL cost is assured from the NHIH budget.

Mortality data quantification was performed using the years of potential productive life lost (YPPLL) and the cost of productivity loss (CPL) indicators [19]. YPPLL is an impact indicator that measures the socio-economic burden of premature deaths, thus estimating the average period the person would have experienced if he had not died due to illness or comorbidity [20].

We used data provided by the National Institute of Statistics (NIS) regarding the percentage of deaths, by year and age category, in the general population. We then kept the same percentage, applied to the number of deaths (ND) associated with depression or comorbid depression, by years and age categories, provided by the NHIH. When calculating YPPLL, we used 9 age classes of 5 years each, from 18 to 63 years for women and 65 for men, as limits for the productive life period. We have thus calculated the time interval lost between the moment of death (considered the median year of the specific age group) and the maximum limit set of the productive life interval. We multiplied the obtained YPPLL with the gross national product (GNP) per capita of the year in which the death occurred, which is supposed to remain constant throughout the entire productive life of the individual. The cost of productivity loss was calculated using the formula:$$CPL=\sum_{i=1}^{9}{ND}_{i}{\times YPPLL}_{i}\times GNP$$
where

*CPL*—cost of productivity loss;

*ND_i_*—number of deaths associated with depression or comorbid depression for each age class of 5 years (*i*);

*YPPLL_i_*—years of potential productive life lost for each age class of 5 years (*i*);

*GNP*—gross national product per capita of the year when death occurred.

### 2.5. Statistical Analysis

For the statistical analysis of the mean differences between the variables, the Student’s *t*-test was used, using the GraphPad software (Addinsoft Software 9, Inc., San Diego, CA, USA) [21]. For gender variable, to compare multiple proportion, K proportion test with Marascuilo procedure was used. Descriptive statistics was performed for all continuous variables (Appendix A).

We used the principal component analysis (PCA) to achieve the following: (i) highlight the correlations between the considered variables, (ii) infer the similarities, respectively, the differences between the statistical units (years) considered by all the recorded variables, and (iii) explain the similarities, respectively the differences between individuals. For this purpose, the results obtained for the statistical units (years) are associated with the results obtained for the statistical variables. For this purpose, XLSTAT software for Excel 2021 [22] was used. The value of *p* < 0.05 was considered statistically significant.

The correlation matrix shows the values of the Pearson’s correlation coefficients between variables, considered two by two. This coefficient analysis allows the study of the link intensity and the evaluation of the possibility of applying the analysis of the main components. High values of these coefficients (over +0.7 or less than −0.7) show a strong correlation between the considered variables. If the value of these coefficients is positive, the links are direct, and if it is negative, the links are inverse.

## 3. Results

### 3.1. Descriptive Analysis

The patient cases analyzed between 2015 and 2021 comprised 2,540,792 patient cases: 1,607,957 (63.29%) were women and 932,835 (36.71%) were men. Women were predominantly affected by depression in a ratio of 1.7 compared to men (Figure 1).

The mean prevalence of depression in the general population was 2.13% ± 0.33 (min 1.63, max 2.56), with a mean prevalence of 1.35% ± 0.21 in women (min 1.02, max 1.65) relative to the mean prevalence of depression in men 0.78% ± 0.11 (min 0.60 and max 0.93).

The main components of the direct cost are the cost of hospitalization and the cost of medication (Table 1 and Figure 2). The direct annual costs for the treatment of depression per patient ranged from a max of 42.45% and min of 27.93%, the medication weighing second in the direct cost.

Hospitalization costs are predominant within the DCs, covering more than half of the spending between 2017 and 2019. However, the constant decrease from the last years placed it at 32.97% in 2021. The total cost of specialized services increased by over 10% from 2015, reaching 18.87% in 2021 (Appendix A). The cost of psychotherapy services also increased in 2021, reaching 6.24% and being relatively double the percentage of other years; yet, the cost of primary care remained consistent, below 1%, throughout the study.

The mean direct cost was EUR 51,068,834 with a mean annual direct cost of depression per patient of EUR 143.

The mean cost of sick leaves was EUR 523,173,605 with a mean sick leaves cost per patient/year with EUR 273 significantly higher than the mean direct cost (*p* = 0.001, *t* = 4.340) (Figure 3).

The invalidity cost was 23.59% of the total in 2015, 39.19% in 2016, 20.08% in 2017, 34.22% in 2018, 32.83% in 2019, 32.48% in 2020, and 9.15% in 2021 (Figure 4). The number of deaths recorded in the adult population is increasing along with the prevalence of illnesses. The CPL value represented 73.00% in 2015; unfortunately, in 2021, it increased to 89.16% (Figure 4).

The mean total cost of depression (direct cost and indirect cost) was EUR 2,015,731,285.86 with the annual mean cost per patient during the study period EUR 5553 for an average of 362,000 patients/year, of which 2.83% corresponded to direct health costs and 97.17% to indirect costs.

### 3.2. Correlations

Data analysis from the correlation matrix (additional word file from Appendix A) showed that:

The number of days of hospitalization is negatively correlated with the number of specialty outpatient consultations (*r* = −0.783, *p* = 0.037) and the number of psychotherapy services (*r* = −0.783, *p* = 0.037).

The cost of hospitalization is negatively correlated only with the number of services in primary care (*r* = −0.799, *p* = 0.03), and there are no other correlations with other analyzed variables.

A strong positive correlation (*r* > 0.974, *p* < 0.001) occurs between the number of psychotherapy services and number of specialized outpatient clinic services.

The cost of psychotherapy is positively correlated (*r* > 0.98 and *p* < 0.001) with the cost of specialized outpatient services (psychiatry) and negatively with the number of days of hospitalization (*r* = −0.762, *p* = 0.03).

The number of medical prescriptions is positively correlated (*r* > 0.97, *p* < 0.001) with the ones of outpatient consultations and psychotherapy services.

The cost of primary care is negatively correlated with the number of days of hospitalization (*r* = −0.833, *p* = 0.02).

The number of sick leaves is not correlated with any other studied variable.

The cost of medication and sick leaves are not correlated with any other variable analyzed.

#### Principal Component Analysis

Principal component analysis (PCA) is an exploratory method of data analysis applied in the study of the relationship between numerical variables. Due to the different nature of the variables used, the data are standardized for the calculation of the distance between two point-values using Euclidean distance. In this study, PCA was used to highlight correlations between the variables considered (shown in Appendix A) and to observe similarities and differences between statistical units. PCA involves the formation of factor axes (principal components) which represent a linear combination of variables that are correlated with each other. These axes make it possible to explain the similarities and differences between the statistical units in terms of all the variables considered.

Since the first two factorial axes cumulate 80% of the total variance (Table 2), our analysis will relate only to them, as shown in Figure 5. The first factorial axis highlights a strong correlation between the variables: number of medical prescriptions consultations, number of prescriptions, outpatient cost, cost of psychotherapy, number of psychotherapy consultations, number of outpatient consultations, and cost of general practice. These variables form a cluster. All these variables are far from the number of consultations during hospitalization, with which they are in a reverse connection.

In order to identify the periods of the different evolution of the values recorded by statistical variables, we made a graphical representation of the points defined by observed years. The diagram shown in Figure 6 reveals three clusters with specific characteristics:−the years 2015 and 2016, located on the same side relative to the first factorial axis, are characterized by the same pattern;−the years 2017, 2018, and 2019 present values closest to the analyzed period’s mean levels;−2020 and 2021, marked by the COVID-19 pandemic, have similar values for all the analyzed indicators.

## 4. Discussion

Mental illness is an individual experience, the responsibility in care belongs to the medical sector, but it also needs attention from society and government policies. Depression is a disorder, but it also manifests as a comorbidity of other chronic diseases, with implications on medical costs, patient’s quality of life, and economic and social costs in general [23]. The analyses of the cost of the disease, along with the cost-effectiveness analysis or other types of evaluations, are necessary for the government policies and have implications on strategic, preventive, and clinical decisions centered on the person [24].

Mental health policies are insufficiently planned, monitored, and reported; thus, policies must be designed with stakeholders and practitioners. They should present strategies for implementation, measuring feasibility, cost-effectiveness, and impact on health outcomes [25].

The current barriers are mainly gravitating around limits of investing in mental health, low political priority, absence of needs-based policy, diagnosis of health disorders, lack of resources, limitation of accessibility due to financial difficulties, and dysfunction in optimizing available resources [26].

Eurostat indicates a depression prevalence in Romania of 1% for 2019, with a European mean prevalence of 7% [27]. In the current study, the mean depression prevalence is 2.13%.

We mention that the prevalence obtained in this study reflects only the patient cases diagnosed and reimbursed in the public health insurance system. People who received a wrong diagnosis, who turned to the private health service system, etc., were not included.

This indicator is increasing in Romania, with mainly women being affected—a trend that has been constantly maintained during the studied period. This result is similar to Shoukai’s findings in a global study on inequalities in mental health. He showed that disparities (i.e., women are twice as likely as men to suffer from mental health illness) have not yet been reflected in health policies [28]. 

*Direct cost*. The results of the present study show that the mean annual direct cost of depression per patient was EUR 142.3, representing half the European mean. The total annual cost of depression in Europe was estimated at EUR 118 billion in 2004, corresponding to EUR 253 per inhabitant [13].

The main components of direct cost are the cost of hospitalization with 36% mean/year and medication with 42% mean/year. The hospitalization cost is predominant in the direct cost, with an oscillation finally leading to a decrease in the final years of the study.

Proportionally opposite, the outpatient services increased as well as the psychotherapy services. There is a tendency for patients with depression to be diagnosed and treated mainly in outpatients. The increased number of outpatient consultations and psychotherapy services caused a substantial decrease in the days of hospitalization. These correlations confirm the results of previous studies conducted around the globe, which demonstrate that psychotherapy and individualized cognitive behavioral therapy (CBT) are more cost-effective than regular healthcare, both alone and in combination with medication [24,29,30].

The mean annual total costs per patient ranged worldwide between USD 1300 to USD 2700, with hospitalization expenses being the significant component in direct costs, as shown in a systematic review conducted by Luppa [17]. Nevertheless, in the present study, the mean annual total costs per patient are EUR 5553, thus much higher.

*Primary medicine* holds less than 1% of the direct cost; it ensures the monitoring and prescription of drugs under the recommendation of psychiatrists (medical letters with a validity of 3 to 6 months). Primary medicine does not offer diagnostic or preventive services; thus, several patients remain unidentified or are treated for other comorbidities, therefore increasing the number of days of hospitalization. The patients directly access the specialty outpatient and are sent to the hospital from these medical services. Therefore, this previously mentioned situation could be another hypothesis.

Direct costs are underestimated due to insufficient knowledge and stigma. Addressability in the healthcare system is conducted either for somatic symptoms or frequent comorbidities of depression. The most known are diabetes mellitus, high blood pressure, inflammatory bowel diseases, and other chronic illness [31,32,33,34].

A meaningful way to stop this cost explosion is by increasing the research in field progress. Moreover, better detection, prevention, treatment, and patient management are imperative to reduce the burden of depression and its costs [13].

*Indirect costs*. The report on the global burden of the disease states that mental disorders account for 13% of all disability-adjusted life years, with years living with disabilities and depression being the leading cause [35]. The indirect cost of sick leave is a burden at present with a mean EUR 273 per patient/year and EUR 523,173,000 annual cost and they are in a ratio of 6:1 compared to CD.

*The morbidity costs* related to lost working days paid through sick leave are EUR 273 per patient annually. The indirect cost of sick leave is a burden at present.

One hypothesis for the high rate of sick leave prescriptions could be that several patients receive them successively without improving their health status. Another hypothesis is that sick leaves are prescribed for long periods of hospitalization.

A concept that can be useful in the discussion is inequity in the use of resources. Morris defines horizontal inequity as the use of different amounts of care for the same needs. He found that people with low incomes and ethnic minorities are more likely to access medical services and that economic status has an effect on the demand for health services [36].

*Mortality associated with depression.* The number of deaths recorded in the adult population is increasing, given the increased prevalence of illnesses.

The mean cost of premature death was EUR 1,441,488,847, and they are in a ratio of 13:1 compared to CD. It is an alarming report; the losses we register as a society are much higher than the costs of care or prevention.

The mean annual total costs per patient ranged worldwide between USD 1300 to USD 2700, with hospitalization expenses being the significant component in direct costs, as shown in a systematic review conducted by Luppa [Eroare!Fărăsursădereferință.]. Nevertheless, in the present study, the mean annual total costs per patient are EUR 5553, thus much higher.


*Clusters of the years*


One direction of analysis can be outlined considering the introduction of a health insurance card for persons and a card reader for the health staff which allows the validation for the consultation in 2015. In the first years, there were difficulties in distributing these to the insured person and in the function of the electronic system that would allow electronic validation of consultation [37].

Another dynamic that we were able to analyze was the regulations on psychotherapy services which are considered auxiliary services, which means the therapist does not have a direct contract with the public health system but through another provider who can prescribe, under certain conditions, psychotherapy services and receive their reimbursement from the public insurance system.

In 2012, in the Framework contract, psychotherapy consultations were regulations only for psychiatrists and speech therapists. In 2014, the specialties that could prescribe were psychiatry, neurology, and otorhinolaryngology. In 2017, the list of prescribers of psychotherapy services was extended to nine specialties. In 2023, the list of prescribers is increased to 23 specialties [38,39,40,41].

*COVID-19 implications*. We face a particular situation during the pandemic years. Although the number of patients with depression and the associated costs increased before 2020, both indicators sharply diminished in 2020 and 2021. The situation was most likely caused by the pandemic restrictions, with limited hospital access and outpatient diagnosis and treatment for any other patient except COVID-19.


*Limitations*


The present study is the first study in Romania that analyzes the cost of depression, according to the data known to us, which, however, has limitations that must be mentioned to support the analysis and correct use of these data.

Uninsured patients and those who benefited from paid services in the private system were not included. Only the public costs insured by the public health insurance system were included. Incorrect coding could be a limitation. Presenteeism, i.e., the decrease in productivity at work, part of the cost of morbidity, included in the indirect cost, was not considered. Additionally, comorbidities state and medication were not included. The socio-demographic variables were not included; yet, they would have been useful in understanding the background of people.

This study did not include additional costs related to suffering, loss of opportunities in education, and reduced participation in family life that could not be quantified [24].


*Solutions*


This analysis is just the tip of the iceberg in solving the burden that depression represents. We only analyzed public health costs. We must not lose sight of the wider context in which these things happen. To find solutions, an integrated approach must be used in which health is alongside economic, social, political, and environmental factors. The perspective of social epidemiology can provide an in-depth understanding of the context and some solutions. Addressing discrimination and inequalities in the social environment can be possible through programs adjusted to the communities and the environment in which they live [42].

Additionally, mental health can benefit from efforts in other areas, such as fighting HIV, improving maternal health, and reducing child mortality. It could access political spaces and become global health priorities, channeling resources, some of which integrate into primary health services [43]. Guidelines for the management of depression should be co-designed by stakeholders and measure feasibility, cost–effectiveness, and impact on health outcomes [25].

Routine monitoring at the patient level has an essential role in individual care and leads to increasing the performance of the mental health care system [44]. A proposal for the primary care system includes depression screening in the annual review and periodic depression screening, especially for patients with chronic diseases.

Moreover, an integrated approach involving a clinical guide to psychological services, including digital health interventions, is considered a prerequisite for increasing the accessibility and effectiveness of mental health services [45]. Another proposal would be to prescribe psychotherapy from the initial diagnosis of the depressive episode, at the first hospitalization, and in treating any chronic disease. Funds for psychotherapy programs, primary care, or outpatient can be reallocated from the economy made to the fund currently allocated to sick leaves or hospitalization. Thus, the cost could turn from a burden into a resource. Experience in this regard already exists. For example, a program for Improving Access to Psychological Therapies was put into place in the UK [46].

## 5. Conclusions

Depressive disorders are a burden for Romanian society. Annual costs represent approximately EUR 2,015,731,285.86 with an annual cost per patient of EUR 5553. Indirect costs (cost of disability and lost productive years) represent 97.17% of the total cost. It is helpful to include depression screening in the annual review and monitor patients with depression in the primary care system. Psychotherapy can play a considerable role. An integrated approach to early diagnosis, effective treatment, monitoring, and prevention as well as included economic and social programs are needed to optimize indirect costs.

## Figures and Tables

**Figure 1 healthcare-11-00921-f001:**
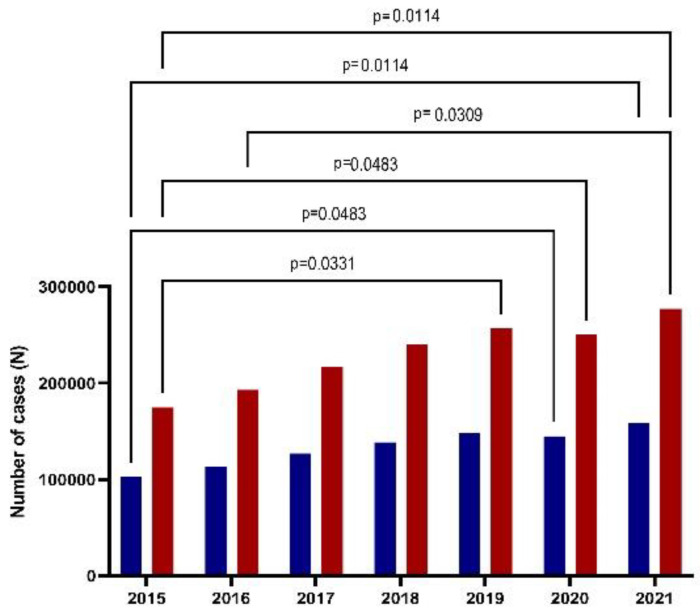
The dynamics of depression cases among years of study (2015–2021) for males (blue) and females (red). (Marascuilo procedure with semnificative critical values).

**Figure 2 healthcare-11-00921-f002:**
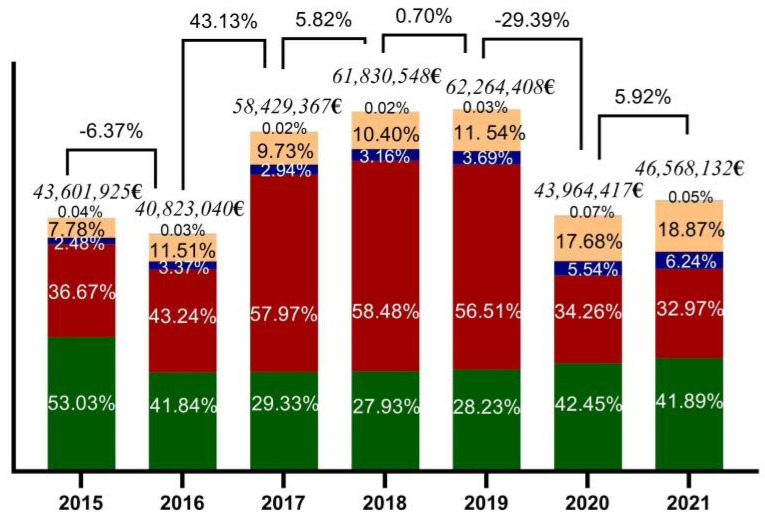
Direct costs in the treatment of depression, compared by year, for primary care (black), specialized services (orange), psychotherapy (dark blue), hospitalization (red), and medication (green).

**Figure 3 healthcare-11-00921-f003:**
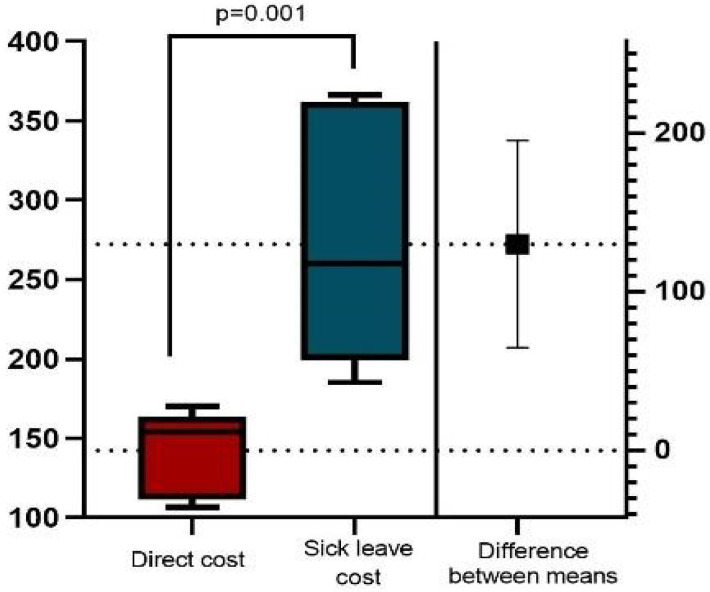
Mean annual patient cost (€) (estimation plot and *p*-value of *t*-test).

**Figure 4 healthcare-11-00921-f004:**
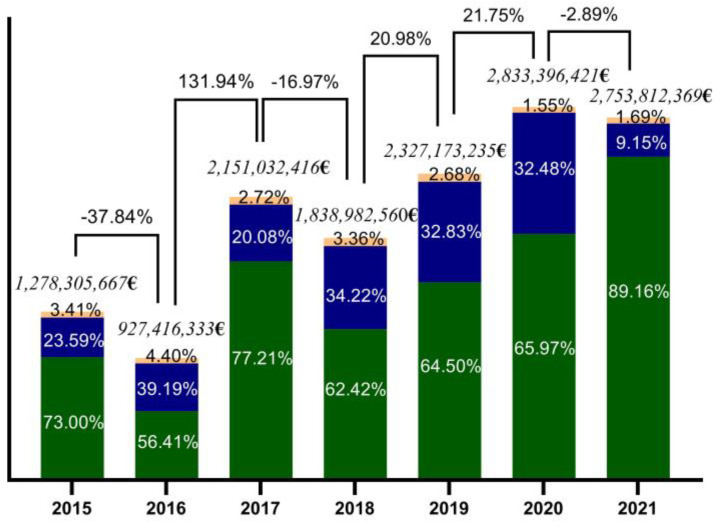
Dynamics of the total cost of depression (direct costs—orange, sick leaves—blue, and CPL—green) between 2015 and 2021 for adult Romanian patient cases; CPL—the cost of productivity loss.

**Figure 5 healthcare-11-00921-f005:**
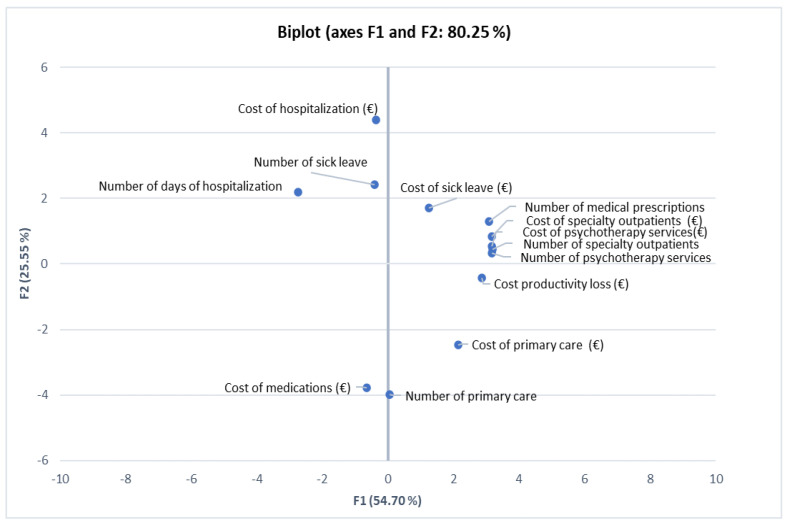
Representation of variables in the system of the first two factorial axes.

**Figure 6 healthcare-11-00921-f006:**
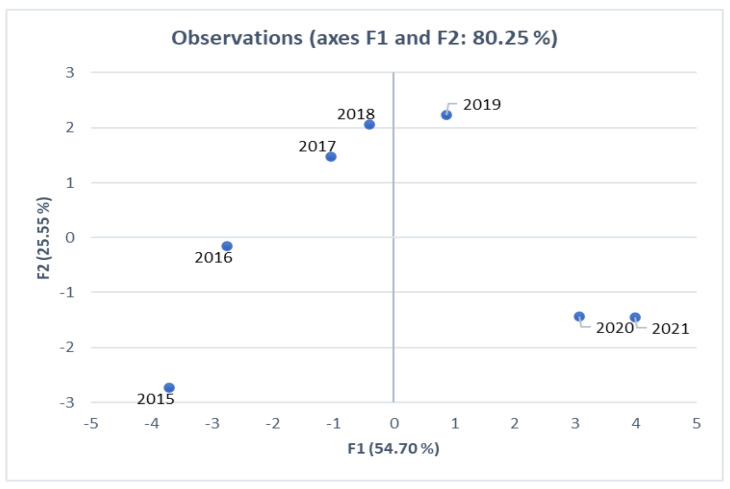
Representation of the years analyzed in the system of the first two factorial axes.

**Table 1 healthcare-11-00921-t001:** Depression costs for patients in Romania between 2015 and 2021.

Number of cases ≥18 years	2,540,792		
Number of deaths ≥18 years	143,340	5.64%	
Depression costs EUR (2015–2021)	Mean/year patient	Cost components	%
**Direct costs**	**142.30**	**357,481,837**	**2.83**
Primary care	0.05	131,420	0.04
Specialty outpatients	16.96	43,953,075	12.30
Psychotherapy	5.42	13,767,622	3.85
Hospitalization	67.12	169,274,345	47.35
Medications	52.88	130,355,375	36.46
**Indirect costs**		**13,752,637,164**	**97.17**
Sick leaves	272.33	3,662,215,236	27.36
Cost of productivity loss (CPL)	-	10,090,421,928	69.81
**Total costs**	**5553.4**	**14,110,119,001**	-

**Table 2 healthcare-11-00921-t002:** Total variance explained.

Component	Initial Eigenvalues			Extraction Sums of Squared Loadings		
	Total	% of Variance	Cumulative %	Total	% of Variance	Cumulative %
1	7.112	54.707	54.707	7.112	54.707	54.707
2	3.322	25.552	80.259	3.322	25.552	80.259

## Data Availability

On demand.

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
