# Peer review of "Depression in Central and Eastern Europe: How Much It Costs? Cost of Depression in Romania"

_healthcare, 2023, doi:10.3390/healthcare11060921_

Round 1
Reviewer 1 Report
In the manuscript, Dr.Lordache and other co-authors conducted an retrospective study to estimate the financial cost of depression in Romania during a 7-years time span. The authors compared components of direct and indirect costs, as well as investigated correlations of these items using principal component analysis. Several suggestions were proposed to use funds efficiently. However, the introduction, methods, and results are questionable and insufficient. And the manuscript is not well-written with low quality. I have multiple major concerns are as follows:
1. The introduction is too short and lack of essential information and detailed description
a. The authors need to provide a comprehensive literature review of current depression in the world or Europe and the depression student program mentioned in line 41.
b. Paragraph about the study aim (lines 46-47) should be put on the end of the introduction
c. The authors need to check the grammars and sentences in the introduction to ensure that precise information is delivered. For example, the first sentence (lines 36-37) and the sentence in line 40-41 need to be rephrased. Please check your language and grammar throughout the article.
2. The Materials and Methods are not well-organized and lack of essential information, which leads to difficulty for reader to understand and capture useful insights.
a. Please give detailed description of the diagnostic method of depression (ICD 10).
b. Please include a subsection in the Methods to introduce the sample size and participants.
c. In the Section 2.3.1 Morbidity cost, the authors need to give a detailed description of how CPL indicators mean and how to calculate it, as for the YPPLL in paragraphs of lines 111-119.
d. In Sectional 2.4 Statistical analysis, please make sure where the Turkey post hoc test was used. In the Results, the authors only addressed the multiple comparison issue in terms of number of depression cases for males and females in 2015-2021. Turkey test is not used for other variables listed in the article.
e. The description of PCA in lines 125-130 are confusing. Please rewrite it and focus on the usage and application of PCA in correlation matrix.
3. The Results are not well-organized, and Tables and graphs were of poor quality.
a. Table 1 needs to be remade and improved. First, text in first column needs to be aligned and bullets need to be removed. Second, the authors should consider using million as the unit to avoid showing very large numbers, or using mean type of cost per person to reflect the amounts. Third, for direct and indirect costs, please list percentages for each subitem.
b. The authors should refer to the first table in the supplemental materials when mentioning costs and percentage by years.
c. Figure 2 and 4 are in low resolution. Please remake them to make numbers more clear
d. When interpreting results from Tables and Figures, the authors should merge related results into one paragraph. Please considering minimizing numbers of paragraphs, especially for Results.
e. Numbers for initial eigenvalues and extraction sums of squared loadings in Table 2 are exactly the same. Please review it and make necessary changes if there are any mistakes.
f. The authors presented some interesting results regarding the interpretation of correlations based on PCA in Figure 5 and 6, especially multiple cluster detection of the years. However, the authors failed to present justification and interpretation about this interesting finding about year trend. Please considering providing some explanation and literature demonstration in the Discussion.
4. In addition to the above major concerns, there are quite a bit grammatical errors and ambiguity throughout the manuscript. A careful revision is recommended to improve the quality of the article addressing the minor concerns listed below.
a. Please be cautious when using the word significant (e.g. line 136) unless p-value is less then 0.05.
b. When reporting numbers in the text (e.g. line 149-151), please use mean(SD, range).
c. When reporting correlation coefficients, please specify which correlation (e.g. Pearson, Spearman, etc) is used.
Please go through the manuscript and revise as much as possible. Improving the quality of the exposition can make an interesting study appealing to more readers.
Author Response
Dear reviewer,
We highly appreciated the constructive comments received that helped us improve our manuscript. We take the opportunity to thank you for your effort!
As suggested, we reviewed the whole manuscript in order to correct the existing errors, improve the structure of the paper and offer a stronger reference list of relevant studies linked to our topic.
Please find below the point-by-point answers to the comments received.
Kind regards,

Reviewer 2 Report
Dear authors
Thank you for your interesting contribution. I have only minor suggestions for improving the paper. However, I will raise some concerns, that can not be solved in this paper.
I would make sure, that the reader understands, that your data only includes diagnosed depression. The indirect costs of depression must be way higher (privately paid therapy, drugs, domestic compensation, misdiagnosed, untreated, and many many more. Maybe you can use the term: public costs.
I would appreciate it, if you include some more critical thinking beyond the mainstream clinical approaches, in your discussion. Critical scholarship has, for a long time, warned about ill-defined societal constitutions, that are now individualized and medicalized under principles of efficacy and for sustaining the system. I value an overview of public costs, but this frame is needed so that readers understand a bit more of the context and meaning and the subject's stance.
Kind Regards.
Author Response
Dear reviewer,
We highly appreciate the constructive comments received that helped us improve our manuscript. We take this opportunity to thank you for your effort!
As you suggested, we have revised the entire manuscript to improve the structure of the paper and provide a stronger perspective.
We completely agreed with your recommendation to use "public cost" to emphasize that we only consider the costs of the public health system. Reframing them with critical thinking is a topic to reflect on. We have used it now and we will use it in future research.
Kind regards,

Reviewer 3 Report
The authors estimated the cost of depression in Romania during the period 2015-2021 using the data obtained from the country’s National Health Insurance System. The manuscript was generally well written. It is publishable subject to some minor revisions and clarifications.
1st Comment: Line 178: The authors stated “The depression total cost range was Euro 927,416,333 to Euro 2,833,396,421…” However, the cost of depression is equivalent to the sum of direct costs, sick leaves, and CPL (indicated in the label of Figure 4; line 182), and the minimum values of direct costs, sick leaves, and CPL are 42,400,544.92, 289,312,047.85, and 850,299,863.42, respectively (given in Table 1). Thus, the minimum total cost of depression in the study should be 1,182,012,456.19 (=42,400,544.92 + 289,312,047.85 + 850,299,863.42) [but not 927,416,333]. So, please your calculations again and figure out when happened.
2nd Comment: Line 37: The first statement of Introduction “…mental health as a way people can be active and creative and citizens involved [1]” is confusing. Probably, it shall be simplified as “…mental health as a way people can be active and creative [1]”.
3rd Comment: Line 141: The authors stated “…comprised 2,540,792 patients…” Please check whether they should be “patient cases” as the label of vertical axis in Figure 1 is “number of cases.”
4th Comment: Line 331: The authors indicated that “Annual costs represent approximately Euro2,138,635,556…” in Conclusions. However, there was no explanation how this figure was obtained. Please clarify.
5nd Comment: There were some minor typing errors such as:
- Line 37: Add a full stop after [1].
- Line 47: “…timespan…” should be “…time span…”
- Line 95: “…[14.” should be “…[14].”
- Line 150: Add a comma after +/- 0.21.
- Line 151: Add a comma after +/- 0.11.
- Line 152: “Error! Reference source not found.” should be “Table 1.”
- Line 162: “Euro42.3…” should be “Euro142.3…”
- Line 170: “…difference between means +/- 29.95” should be “…difference between means 130.03 +/- 29.95.” (Please check the mean values using the collected data).
- Line 258: “…depression per patient was Euro 134…” should be “…depression per patient was Euro 142.3…” (see Table 1).
- Line 317: What is “Error! Reference source not found.”?
- Line 322: “…[36,36]” should be “…[36,37].”
- Lines 352-353: Please delete these sentences: “All appendix sections must be cited in the main text…”
- Line 356: “…-our-respons” should be “…-our-response.”
- Line 359: The volume, issue, and page numbers of Chisholm’s (2003) paper should be “183(2), 121-131”.
- Line 367: “…The. British…” should be “…The British” i.e. removing the full stop between “The” and “British”.
- Line 371: “Journal of afective disorder…” should be “Journal of Affective Disorder…”
- Line 377: “Organization., W.H….” should be “World Health Organization”
- Lines 382-383: This reference should be “Rumisha SF, George J, Bwana VM, Mboera LE. Years of potential life lost and productivity costs due to premature mortality from six priority diseases in Tanzania, 2006-2015. PLoS One (2020), 15(6), e0234300. -
- The volume, issue, and page numbers of Katon’s (2022) paper should be “13(1), 7-23”.
- Lines 412-413: This reference i.e. [32] is identical to [19]. So, please delete it and renumber the rest of references in the text and the list of References.
- Lines 426-427: Delete “Centre for Economic Performance, LSE.” Additionally, this reference should be “Clark DM, Canvin L, Green J, Layard R, Pilling S, Janecka M. Transparency about the outcomes of mental health services (IAPT approach): an analysis of public data. The Lancet (2018), 391(10121), 679-686.”
- Lines 428-431: I could not find these two references in the text. Please check carefully.
Author Response
Dear reviewer,
We highly appreciated the constructive comments received that helped us improve our manuscript. We take the opportunity to thank you for your effort!
As suggested, we reviewed the whole manuscript to correct the existing errors and improve the paper.
Please find below the point-by-point answers to the comments received.

Round 2
Reviewer 1 Report
This revision looks reasonable and successfully addresses most of my review comments.